# Evaluation of Anti-Hyperglycemia and Complications of Red and Black Thai Jasmine Rice Cultivars in Streptozotocin-Induced Diabetic Rats

**DOI:** 10.3390/molecules27228043

**Published:** 2022-11-19

**Authors:** Nittiya Suwannasom, Chutamas Thepmalee, Krissana Khoothiam, Chonthida Thephinlap

**Affiliations:** 1Division of Biochemistry, School of Medical Sciences, University of Phayao, Phayao 56000, Thailand; 2Unit of Excellence of Research and Development for Cancer Therapy, School of Medical Sciences, University of Phayao, Phayao 56000, Thailand; 3Division of Microbiology, School of Medical Sciences, University of Phayao, Phayao 56000, Thailand

**Keywords:** black rice, red rice, hyperglycemia, diabetic rats, phytochemical analysis

## Abstract

The phytochemical constituents of red (RR) and black (BR) rice extracts were determined using high-pressure liquid chromatography (HPLC). Phytochemical screening revealed the presence of catechin, rutin, isoquercetin, cyanidin 3-glucoside, cyanidin 3-O-rutinoside, peonidin and quercetin. The anti-diabetic activities of RR and BR extracts on diabetic complications were examined in a streptozotocin-induced diabetic rat model. Rats (n = 80) were divided into 10 groups (n = 8 rats per group). Healthy and diabetic RR or BR-treated groups received 10, 50, or 200 mg of RR or BR per kg of body weight daily for 45 days. The results demonstrated significantly improved glucose control in rats administered RR or BR, while triglyceride and cholesterol levels were reduced in the diabetic groups. Moreover, RR or BR treatment led to decreased levels of malondialdehyde, aspartate aminotransferase, alanine aminotransferase, blood urea nitrogen, and creatinine. Further, glutathione concentration was significantly increased in both serum and liver tissue from RR- and BR-treated diabetic rats.

## 1. Introduction

Diabetes mellitus (DM) is caused by impaired insulin secretion, insulin resistance, or both and leads to high blood glucose levels (hyperglycemia) [1]. Diabetes is a public health issue that is increasing in prevalence; In 2021, approximately 537 million people worldwide were living with diabetes, and the number is predicted to climb to 643 million in 2030 [2]. Further, DM caused 6.7 million deaths in 2021, which were related to diabetes and its various complications, which include retinopathy, nephropathy, and neuropathy [3].

Chronic hyperglycemia leads to the development of oxidative stress in diabetes due to multiple factors, including imbalanced cellular redox processes, reduced antioxidant defenses, free radicals generated by glucose autoxidation, and elevated levels of some pro-oxidants, such as ferritin and homocysteine. Increased formation of intracellular advanced glycation end-products (AGEs) also has a vital role. Under hyperglycemic conditions, glucose reacts with specific amino acid residues of proteins to form a modification known as glycation. Glycation of proteins leading to AGEs can contribute to alterations in cell signaling and free radical production [4,5] and various consequent diabetic complications [6].

Notably, global healthcare expenditure for the management of diabetes and its complications is expected to reach USD 578 million in 2030 [2]. Hence, control of diabetes, as one of the fastest-growing global health emergencies of the 21st century, is essential to reduce health spending on treatment and prevention of complications. Plant medicines, as rich sources of natural antioxidants, can have positive effects on metabolic disorders such as DM [7,8]. Further, routine dietary sources of natural antioxidants are very important, as they are generally easily available and suitable for use as dietary interventions. In the present study, we investigated pigmented rice as a routine antioxidant-rich food, which has protective effects against DM and its related complications.

Rice (*Oryza sativa* L.), a member of the grass family (Poaceae), is a major source of carbohydrates for over half of the world’s population. Red and black rice are special rice varieties that are cultivated in different regions of Asia, including Thailand. Colored rice species are well known for their nutritional value and are considered potential sources of antioxidants as well as functional foods [9,10]. Antioxidant activities and phytochemical constituents vary greatly among different black and red rice varieties [11,12]. Thai black rice has a high anthocyanin profile, as well as substantial phenolic and flavonoid contents, and is well known for its ability to scavenge free radicals relative to other Thai rice cultivars [12,13,14]. There have been numerous in vitro studies evaluating the antioxidant and anti-inflammatory properties of rice-derived polyphenols [15]. Potential anti-inflammatory mechanisms of action of colored rice-derived polyphenols show that polyphenols may inhibit the mitogen-activated protein kinase signaling cascade [16]. Previous reports have noted the anti-oxidative effects of pigmented rice consumption on diseases influenced by oxidative stress, such as diabetes [17,18]. 

Recently, the Thai red rice cultivar Hom Dang Sukhothai 1 rice extract was shown to have high anthocyanin, phenolic, and flavonoid contents with high in vitro antioxidant potential [19,20]. A previous report also showed that the Thai black rice cultivar, Hom Dum Sukhothai 2, has high phenolic, flavonoid, anthocyanin, and tannin contents [19]. However, Hom Dang Sukhothai 1 and Hom Dum Sukhothai 2 have no detailed reports on their anti-diabetic properties to date. Therefore, the antidiabetic activity of these two Thai pigmented rice varieties (Hom Dang Sukhothai 1 red and Hom Dum Sukhothai 2 black jasmine rice cultivars) were analyzed to assess their potential for application in the amelioration of diabetes.

## 2. Results

### 2.1. Phytochemical Screening of RR and BR

HPLC analyses of RR and BR revealed various phytochemical contents, including: catechin, rutin, i-quercetin, cyanidin 3-glucoside (C3G), cyanidin 3-O-rutinoside, and peonidin (total) (Table 1). The content of the main polyphenols was lower in RR than in BR, and anthocyanins were not detected in RR. Among polyphenols, both RR and RR contained the highest levels of rutin (1097.57 and 684.06 mg/100 g extract, respectively). Of the anthocyanins, BR had the highest C3G content (446.30 mg/100 g extract). Detailed chromatograms are presented in Appendix A.

### 2.2. Effects of RR and BR on Rat BW

As shown in Table 2, diabetic control group rats had a significantly lower BW than healthy controls (*p* < 0.01); however, the BW of RR- or BR-treated diabetic rats was significantly higher (*p* < 0.05) than those of the diabetic control group.

### 2.3. Effects of RR and BR on FBG

STZ-induced diabetic rats had significantly higher FBG levels than those of control rats (*p* < 0.01). At the end of 45 days of treatment, FBG did not differ between untreated healthy rats and healthy rats treated with RR or BR (Appendix A); however, FBG was reduced after 15 days of treatment of diabetic rats with both RR and BR at every concentration (*p* < 0.05) (Figure 1).

### 2.4. Effects of RR and BR on Blood TG and CHO Levels

TG levels were significantly higher in serum from rats administered STZ after 7 days than in healthy control rats (*p* < 0.05). TG levels were significantly decreased in diabetic rats treated with 50 and 200 mg/kg BW RR and BR at every concentration in a dose-dependent manner (*p* < 0.05). Further, diabetic group rats treated with 10 mg/kg BW RR had lower TG levels; however, the difference was not significant after 30 days of treatment (Figure 2A). Levels in untreated healthy control rats did not differ significantly from those of RR- and BR-treated healthy controls (Appendix A).

Levels of CHO in diabetic rats were higher than those in healthy control rats (Figure 2B); nevertheless, levels were similar in healthy controls and healthy rats treated with RR or BR (Appendix A). CHO levels were significantly reduced in diabetic rats treated with RR or BR at every concentration for 15 days relative to untreated diabetic control rats (*p* < 0.05). These changes continued and were significant after 45 days in RR- or BR-treated diabetic rats (*p* < 0.05), which had lower CHO levels than those of healthy control animals.

### 2.5. Effects of RR and BR on MDA, FRAP, and GSH Levels

Plasma MDA and FRAP levels were significantly higher in diabetic than healthy control rats (*p* < 0.05) (Figure 3A), while MDA levels were similar in RR- and BR-treated and untreated healthy controls (Appendix A). After treatment with RR and BR at 50 and 200 mg/kg for 45 days in diabetic rats, MDA levels were decreased as compared to those in diabetic control rats (*p* < 0.05); however, no significant differences in plasma FRAP levels were observed among any groups (Figure 3B).

The results showed that STZ-diabetic rats exhibited a highly significant decrease in serum GSH levels relative to healthy controls (*p* < 0.01), while those in groups treated with every concentration of both RR and BR showed a marked elevation relative to untreated diabetic rats (*p* < 0.05) (Figure 4A). Interestingly, treatment of healthy rats with 200 mg/kg RR led to significantly improved GSH levels (*p* < 0.01) relative to control rats (Appendix A).

STZ-diabetic rats had lower GSH levels in liver tissue than healthy control rats, while treatment of diabetic rats with RR or BR (50 and 200 mg/kg) significantly increased (*p* < 0.05) GSH levels in liver tissue relative to untreated diabetic rats in a dose-dependent manner (Figure 4B).

### 2.6. Effects of RR and BR on AST and ALT Levels

Levels of AST and ALT were significantly increased in STZ-induced diabetic rats compared to healthy control animals (*p* < 0.05, 0.01) (Figure 5A,B), while these levels were similar in healthy RR- or BR-treated and untreated rats (Appendix A). A significant reduction in AST levels was observed in diabetic rats after 15 days of administration with 200 mg/kg RR and after 30 days with 50 mg/kg RR, as well as with BR at every concentration after 15 days of administration (*p* < 0.05). 

BR treatment at 200 mg/kg led to a significant reduction of ALT levels after 15 days in diabetic rats relative to diabetic controls (*p* < 0.05) (Figure 5B). Although levels of ALT were slightly reduced compared with untreated diabetic rats after RR treatment, the difference was not significant. Hence, ALT levels were not improved by RR treatment. 

### 2.7. Effects of RR and BR on BUN and Creatinine Levels

Untreated diabetic rats had significantly higher BUN and creatinine levels (*p* < 0.05, 0.01) relative to the healthy control group (Figure 6A,B), while BUN levels did not differ between RR- and BR-treated and untreated healthy rats (Appendix A). In contrast, 15 days of treatment with all doses of RR and BR led to a significant lowering of BUN levels in diabetic rats relative to untreated diabetic control animals (*p* < 0.05) (Figure 6A).

Further, 15 days of administration of every concentration of RR and BR led to significant decreases in creatinine levels (*p* < 0.05) (Figure 6B), which were consistently reduced until the end of the experiment. These data clearly demonstrate that RR and BR administration can reduce creatinine levels. Moreover, creatinine levels did not differ between untreated and RR- or BR-treated healthy rats after 45 days of treatment (Appendix A).

## 3. Discussion

Pigmented rice can be classified as a functional food due to its antioxidant properties, which provide health benefits. Red and black rice varieties have been intensively investigated due to several beneficial biological activities, particularly their antioxidant and anti-inflammation properties. In the present study, phytochemical analysis of RR and BR revealed the presence of various polyphenol compounds, including catechin, rutin, and i-quercetin; however, anthocyanins were undetectable in red rice, as mentioned previously by Pengkumsri et al. These observations are similar to those of previous reports that the black cultivar, Hom Dam Sukhothai 2, possessed higher contents of both total anthocyanins and total phenolics than those of the red rice cultivar, Hom Dang Sukhothai 1. This result is also similar to a previous report showing that black rice has a higher total anthocyanin content than red rice [11,21,22]. The results of the present study indicate that C3G is the major anthocyanin found in BR, with lesser amounts of peonidin, quercetin, and cyanidin 3-O-rutinoside present. It has also been reported that black rice has an approximately 35-fold higher anthocyanin content than that of red rice [21]. 

Diabetes is a metabolic disease characterized by in mpaired ability to produce or respond to insulin that results in hyperglycemia, which can become severe and lead to numerous serious health complications. In the present study, we aimed to examine the effects of RR and BR on STZ-induced diabetes in rats. The diabetogenic agent STZ enters the pancreatic islets via glucose transporter (GLUT)-2 and induces irreversible destruction of pancreatic beta cells, resulting in alkylation of DNA and subsequent cell necrosis [23]. Therefore, STZ can lead to the dysfunction of pancreatic beta cells to synthesize and secrete sufficient insulin, causing physio-metabolic abnormalities and resulting in weight loss with a reduction of glucose, TG, and CHO blood levels.

The reduction in BW in STZ-induced diabetic rats is related to the breakdown of structural proteins and muscle wasting [24]. However, treatment with RR and BR led to weight gain in STZ-induced diabetic rats, with those receiving RR exhibiting a higher weight gain than those treated with BR. These data confirm the benefits of colored rice for weight control [25]. Positive effects on blood glucose levels were also observed in RR- and BR-treated diabetic rats. The biological properties of RR and BR can be attributed, in part, to their phytochemical components. C3G-rich anthocyanin extract, especially from black rice, has hypoglycemic effects on diabetic rats [26,27]. There are several possible mechanisms by which RR and BR supplementation may restore glucose levels. It is possible that C3G can increase insulin sensitivity and up-regulate GLUT4 in adipose tissue, as well as downregulate inflammatory adipocytokines (e.g., TNF-α, IL-6, and MCP-1) in diabetic mice models [28]. C3G also contributes to the expression and translocation of GLUT-1 and GLUT-4 in adipocytes and muscle cells, leading to increased glucose uptake and subsequent reduced plasma glucose levels [29]. Furthermore, the anthocyanin and proanthocyanidin content of black rice and red rice has inhibitory effects on carbolytic enzyme activity, thereby disturbing starch and processes that suppress or delay glucose absorption [30,31]. Nazir et al., (2021) also reported that catechin exhibited strong dose-dependent inhibition of α-amylase and α-glucosidase enzymes [32]. Some polyphenols directly influence cell signaling pathways, and these antioxidant pigments exhibit strong radical scavenging activities and ameliorate hyperglycemia in diabetic mice [29].

In the current study, levels of TG and CHO were reduced in a dose-dependent manner in STZ-induced diabetic rats after treatment with RR and BR. This is consistent with the earlier observation that dietary pigmented rice extract reduces serum TG and total CHO levels [33,34]. The possible mechanisms of pigmented rice anthocyanin action in lowering TG may involve decreasing levels of the primary TG transporters, apolipoprotein B and apolipoprotein C-III-lipoprotein [35]. Furthermore, consumption of polyphenol-rich colored rice has also been proven to increase CHO degradation by elevating cholesterol 7-α-hydroxylase, and sterol 12-α-hydroxylase expression, while lowering hepatic CHO synthesis by inhibiting the activity of two key enzymes, 3-hydroxyl-3-methylglutaryl coenzyme A reductase and acyl-coenzyme A cholesterol acyltransferase-2 [36]. Further, black rice has hypocholesterolemic effects in hypercholesterolemic rats by increasing CHO excretion [37]. Moreover, one anthocyanin in pigmented rice, C3G, acts as an HMG-CoA reductase inhibitor, leading to reduced CHO biosynthesis [38]. In addition, it has been suggested that individual phenolic compounds and combinations of polyphenols can have CHO-lowering effects [36]. 

The interaction of AGEs and receptors for AGEs (RAGEs) elicits oxidative stress generation by activating the NADPH oxidase pathway [39], leading to activation of signaling cascades involved in endothelium damage, which are strongly suggested to be major contributory factors in the development of diabetic complications [40]. AGEs augment ROS formation through the depletion of GSH [41]. Hence, the AGE-RAGE interaction increases levels of NADPH oxidase and decreases levels of GSH. Our study shows that MDA levels are increased, while those of GSH are decreased, in STZ-induced diabetic rats. RR and BR treatments decreased MDA levels while, in contrast, incraseding GSH levels. The bioactive ingredient of RR, C3G, is also involved in AGE-RAGE inhibition by the formation of AGE-C3G-RAGE complexes that regulate cellular pathways involved in pro-inflammatory processes and oxidative damage [42]. Administration of BB or RR inhibits AGE-RAGE formation, leading to decreased NADPH oxidase activity; thus, decreased oxidative stress raises GSH levels. 

Previous studies have reported similar findings, showing that RR and BR extract-treated diabetic rats show a significant reduction in MDA and a significant elevation of GSH in serum and liver tissue [25]. Further, Nizamutdinova et al. reported similar results showing that the administration of anthocyanins in the seed coats of black soybeans resulted in a significant decrease in MDA levels in STZ-induced diabetic rats [43]. Furthermore, administration of catechin reduces MDA levels but elevates GSH-Px activity in cardiac tissue from STZ-induced diabetic rats [32]. These findings suggest that antioxidant-rich pigmented rice extracts may relieve chronic oxidative stress. Since compounds obtained from RR and BR extracts possess potent antioxidant effects, these are likely primarily responsible for the reduction of MDA levels observed in STZ-induced diabetic rats in this study.

The elevation of liver enzyme levels can indicate liver injury and dysfunction. Increased AST and ALT levels were observed in diabetic rats, while administration of RR or BR caused a reduction in the activity of liver enzymes and thus alleviated the liver injury associated with STZ-induced diabetes. This finding corresponds with that of a previous study performed by Almundarij et al. who reported that administration of colored rice extracts led to significant decreases in ALT and AST levels [25].

Furthermore, rises in serum BUN and creatinine levels in STZ-diabetic rats can indicate renal dysfunction with hyperfiltration, characteristic of diabetic nephropathy. Our data showed that although serum creatinine levels in all groups are normal (ranging from 0.5 to 0.8 mg/dL [44], serum creatinine levels in all the rats significantly increased after STZ induction compared to the normal control rats. Therefore, administration of RR or BR caused a decrease in renal function damage and improved renal dysfunction in diabetic rats. Unfortunately, the BUN levels in the treated diabetic group were lower than those in the normal control group at the end of the experimental period. This is consistent with previous reports that have shown that oral administration of mulberry extract containing 40% of C3G at a very high dose of 4.2 g/kg decreased BUN levels in the diabetic group, which were reversed after the recovery period. However, these changes were regarded as having no toxicological significance because they were small and not dose-dependent [45].

These results are consistent with the findings of previous studies in which anthocyanins, particularly C3G, were shown to attenuate liver steatosis and adipose inflammation in diabetic rats [29]. Furthermore, previous studies have reported that black rice can protect the liver from damage, including fatty liver disease, and these hepatoprotective activities have been related to its antioxidant capacity [34,46]. In addition, black rice is reported to increase fatty acid metabolism and reduce the risk of hyperglycemia and cholesterolemia [47].

## 4. Materials and Methods

### 4.1. Plant Material and Preparation of Rice Extracts from Red Rice (RR) and Black Rice (BR) 

The two Thai colored rice cultivars, Hom Dang Sukhothai 1 (PSL96016-B4-3-1-ST-1), and Hom Dum Sukhothai 2 (PSL00284-12-2-5R SKT-1), were kindly supplied by the Organic Agriculture Project, Sukhothai Airport (Sukhothai, Thailand). Dried rice powder (250 g) was extracted with 1 L of 70% (*v*/*v*) ethanol at a shaking rate of 150 rpm for 12 h. Subsequently, mixtures were filtered through Whatman No. 1 filter paper, and solvents were removed using a rotary evaporator at 70 °C under vacuum to produce RR and BR powders, which were stored at −20 °C until use. The chemical constituents of RR and BR extracts were further quantitated by high-performance liquid chromatography (HPLC) at the Central Laboratory Company Service Center (Chiangmai, Thailand). The detailed procedures are mentioned in the Appendix A.

### 4.2. Animals

Healthy Wistar albino rats (120–150 g) were procured from the National Laboratory Animal Centre of Thailand at Mahidol University. During the experiment, rats were housed under a constant temperature (25 ± 2 °C) and humidity (60% ± 5%) with a 12 h dark cycle and were fed with a standard diet (Perfect Companion Co. Ltd., Thailand). All animal procedures were approved by the Animal Ethics Committee at the University of Phayao (Protocol number, NU-AE530718).

### 4.3. Induction of Diabetes

Rats were randomly distributed into ten groups (n = 8 rats per group) as follows: 

Group 1, healthy control rats; 

Group 2, healthy rats receiving 200 mg RR/kg body weight (BW);

Group 3, healthy rats receiving 200 mg BR/kg BW; 

Group 4, diabetic rats; 

Group 5, diabetic rats receiving 10 mg RR/kg BW; 

Group 6, diabetic rats receiving 50 mg RR/kg BW; 

Group 7, diabetic rats receiving 200 mg RR/kg BW; 

Group 8, diabetic rats receiving 10 mg BR/kg BW; 

Group 9, diabetic rats receiving 50 mg BR/kg BW;

Group 10, diabetic rats receiving 200 mg BR/kg BW.

Diabetes was induced by a single intraperitoneal injection of streptozotocin (STZ; Sigma Aldrich, USA) at 75 mg/kg BW in 100 mM of a sodium citrate buffer (pH 4.5). The healthy control group was injected with the sodium citrate buffer alone. After 7 days of induction, rats with fasting blood glucose (FBG) levels of 250 mg/dL were judged to have diabetes and selected for use in further experiments.

### 4.4. Biochemical Analyses 

Rice extracts were administered over a period of 45 days via oral gavage, 7 days after diabetes induction. Rat BW was measured before treatment and then every 15 days. Every 15 days, fasting blood samples were taken from the tail vein before treatment with sodium fluoride to examine blood glucose; or ethylenediaminetetraacetic acid to evaluate cholesterol (CHO), triglyceride (TG), glutathione (GSH), malondialdehyde (MDA), aspartate aminotransferase (AST), alanine aminotransferase (ALT), blood urine nitrogen (BUN), creatinine levels, and ferric ion reducing antioxidant power (FRAP) assay using commercial test kits (Randox Laboratories Limited, Antrim, UK) according to manufacturer’s instructions. At the end of the 45-day experimental period, all rats were sacrificed.

### 4.5. Statistical Analysis

Data were analyzed and compared using a one way ANOVA followed by a Tukey’s post-hoc test. Data are presented as the mean ± the standard error of the mean (S.E.M.).

## 5. Conclusions

The present study showed that administration of RR or BR resulted in significantly reduced blood glucose, CHO, TG, and oxidative stress and improved liver and renal function biomarkers in STZ-induced diabetic rats. Administration of RR or BR also prevented the development of oxidative stress, affecting GSH induction in STZ-induced diabetic rats. Given their nutritional benefits, the use of pigmented rice extracts as food supplements or the consumption of rice as a staple food may prove to have powerful beneficial effects on diabetes-related complications. 

## Figures and Tables

**Figure 1 molecules-27-08043-f001:**
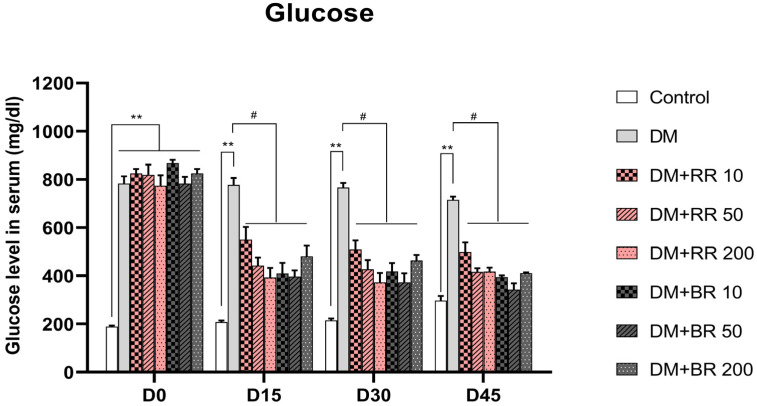
Effects of various doses of RR and BR on blood glucose levels in STZ-induced diabetic rats. ** *p* < 0.01, diabetic control group vs. healthy control group; # *p* < 0.05, treated group vs. diabetic control. Values represent mean ± S.E.M. (n = 8).

**Figure 2 molecules-27-08043-f002:**
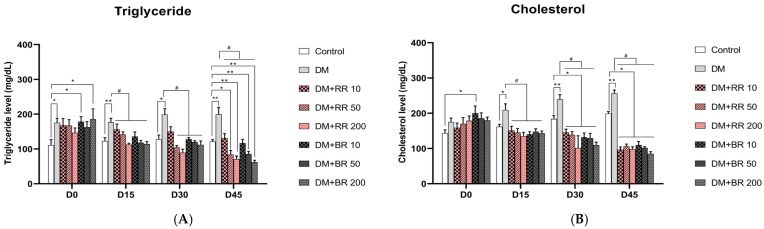
Effects of various doses of RR and BR on triglyceride (**A**) and cholesterol (**B**) levels in STZ-induced diabetic rats. * *p* < 0.05 and ** *p* < 0.01, diabetic control group vs. healthy control group; # *p* < 0.05, treated group vs. diabetic control. Values represent mean ± S.E.M. (n = 8).

**Figure 3 molecules-27-08043-f003:**
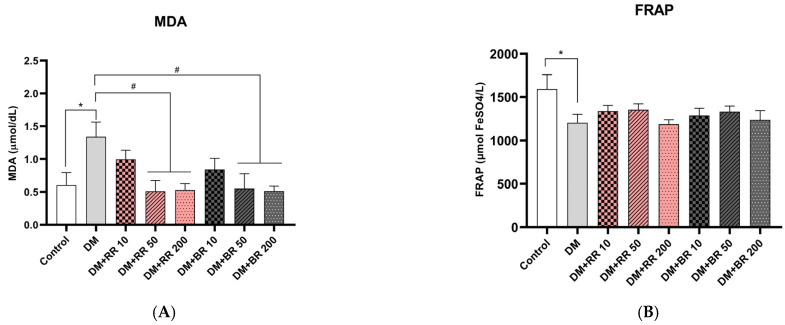
Effects of RR and BR on MDA (**A**) and FRAP (**B**). Values represent mean ± S.E.M. (n = 8). * *p* < 0.05, diabetic control group vs. control group; # *p* < 0.05, treated group vs. diabetic control.

**Figure 4 molecules-27-08043-f004:**
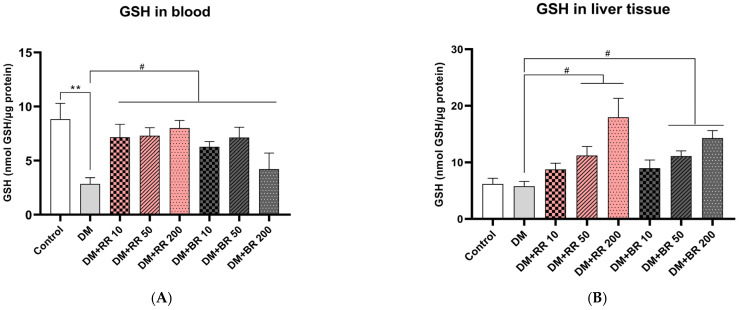
Effects of RR and BR on serum (**A**) and liver tissue (**B**) GSH levels. ** *p* < 0.01, diabetic control group vs. control group; # *p* < 0.05, treated group vs. diabetic control. Values represent mean ± S.E.M. (n = 8).

**Figure 5 molecules-27-08043-f005:**
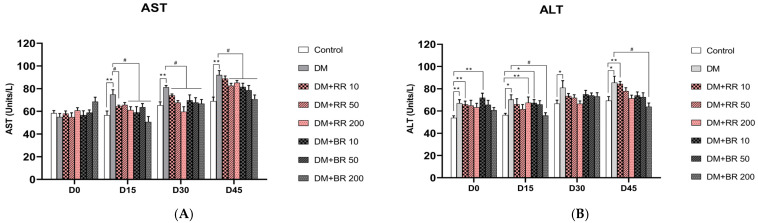
Effect of various doses of RR and BR at on AST (**A**) and ALT (**B**) levels in STZ-induced diabetic rats. * *p* < 0.05 and ** *p* < 0.01, diabetic control group vs. control group; # *p* < 0.05, treated group vs. diabetic control. Values represent mean ± S.E.M. (n = 8).

**Figure 6 molecules-27-08043-f006:**
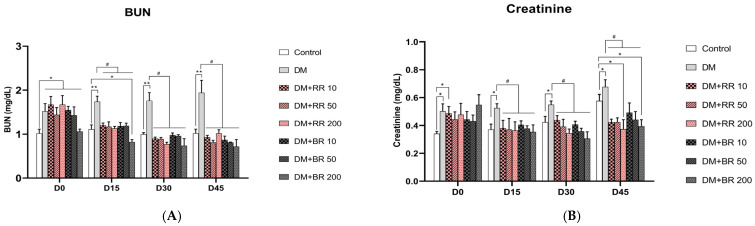
Effects of various doses of RR and BR on BUN (**A**) and creatinine (**B**) levels of STZ-induced diabetic rats. * *p* < 0.05 and ** *p* < 0.01, diabetic control group vs. control group; # *p* < 0.05, treated group vs. diabetic control. Values represent mean ± S.E.M. (n = 8).

**Table 1 molecules-27-08043-t001:** Phytochemical content in RR and BR ethanolic extracts.

Phytochemical Content	(mg/100 g Extract)	
	RR	BB
**Polyphenols**		
Catechin	297.42	342.86
Rutin	1097.57	684.06
Isoquercetin	672.10	430.44
**Anthocyanins**		
cyanidin 3-glucoside	ND	446.30
cyanidin 3-O-rutinoside	ND	24.21
Peonidin	ND	115.35
Quercetin (total)	ND	35.70

ND = Not detectable

**Table 2 molecules-27-08043-t002:** The effects of various doses of RR and BR on STZ-induced diabetic rats body weight.

Group	Body Weight (g)			
	D0	D15	D30	D45
Healthy Control	266.88 ± 3.65	290.63 ± 2.90	328.75 ± 3.75	380.00 ± 5.09
Diabetic control	124.29 ± 3.30 *	113.57 ± 4.65 *	121.43 ± 5.18 *	125.71 ± 5.99 *
Diabetic RR treated group				
10 mg/kg BW	142.50 ± 12.39 *	153.00 ± 17.30 *	157.00 ± 21.05 *	221.67 ± 29.91 *^,#^
50 mg/kg BW	163.57 ± 9.50 *	155.71 ± 11.43 *	156.43 ± 12.78 *^,#^	167.14 ± 13.80 *^,#^
200 mg/kg BW	131.43 ± 8.52 *	123.57 ± 7.75 *	129.17 ± 9.38 *	148.33 ± 14.90 *^,#^
Diabetic BR treated group				
10 mg/kg BW	124.29 ± 6.57 *	120.00 ± 6.42 *	124.17 ± 2.26 *^,#^	120.83 ± 6.26 *
50 mg/kg BW	146.00 ± 9.79 *	141.52 ± 10.98 *	152.96 ± 10.92 *	165.36 ± 17.81 *^,#^
200 mg/kg BW	130.45 ± 5.15 *	139.52 ± 3.71 *	139.76 ± 7.16 *	154.43 ± 10.35 *^,#^

* *p* < 0.01, diabetic control group vs. healthy control group; # *p* < 0.05, treated group vs. diabetic control. Values represent mean ± S.E.M. (n = 8).

## Data Availability

Not applicable.

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
