# Peer review of "Evaluation of Anti-Hyperglycemia and Complications of Red and Black Thai Jasmine Rice Cultivars in Streptozotocin-Induced Diabetic Rats"

_molecules, 2022, doi:10.3390/molecules27228043_

Round 1
Reviewer 1 Report
In this paper, the authors continue their research on the antiglycemic effects of natural extracts. The work, even if not very innovative, is clear in the description.
However the first thing noticed from the SI material in particular by the HPLC attached, is the absence of resolution of the peaks, in particular the spectrum Fig S2 and Fig S3, from which the authors calculate the concentration of the components present in the extract.
My experience leads me to say that from the attached spectrum these calculations have no value. The Fig S5 and Fig S6 spectra are slightly more resolved, here too it would be appropriate to improve the quality of the analysis.
It is necessary to repeat the HPLC analysis in order to have a better resolution so as to make the calculations made reliable.
Please modify appropriately.
- line 79: please clarify the meaning of i-quercetin.
- line 93 : the authors wrote **P, but it is not present in the table.
- line 131: please check the sentence, are there some missing words?
- line 195: However, capital letter
• publish after major revision taking into account the comments above.
Reviewer 2 Report
This paper provide a comparison of the antidiabetic affects of Thai red rice and Thai black rice in a STZ diabetic rat model with diabetic and non diabetic rat control groups. The number of animals in each group is good (n=8), the trial was undertaken at different doses which could reasonably be translated to human use. Both red and black rice treatment reduced blood glucose midway between diabetic and non diabetic controls, although there was not a strong dose response effect. There was some benefit in trigycerides and cholesterol but the modulation of MDA and GSH by both black and red rice brought values back to values in non diabetic controls. There did not appear to be a great difference in the two types of rice.
Generally the pharmacological methods are reasonable and well analysed, although it would be useful to have a drug control such as metformin. The results are promising especially where inhibition of glycation is concerned. The manuscript is well written. One area that needs attention is the phytochemisty. Although standard methods are applied for analysis of anthocynins and other phenolics, the large amount of proanthocyanins has created an analytical problem by producing a large wide peak in the baseline of the chromatograms. This should be acknowledged in the manuscript and further work to identify these difficult to analyse oligomeric compounds should be suggested in the discussion. I suggest that the tannic acid data be deleted because the commercial standard is almost certainly not relevant to your products.
Round 2
Reviewer 1 Report
the authors have made the required changes